

# Buoyant forces promote tidewater glacier iceberg calving through large basal stress concentrations

Matt Trevers[1], Antony J. Payne[1], Stephen L. Cornford[1,2], Twila Moon[3]

[1]Centre for Polar Observation and Modelling, School of Geographical Sciences, University of Bristol, Bristol, BS8 1SS, UK
[2]Department of Geography, Swansea University, Swansea, SA2 8PP, UK
[3]National Snow and Ice Data Center, University of Colorado, Boulder, CO 80309-0449 USA

*Correspondence to*: Matt Trevers (matt.trevers@bristol.ac.uk)

**Abstract.** Iceberg calving parameterisations currently implemented in ice sheet models do not reproduce the full observed range of calving behaviours. For example, though buoyant forces at the ice front are known to trigger full-depth calving events on major Greenland outlet glaciers, a multi-stage iceberg calving event at Jakobshavn Isbræ is unexplained by existing models. To explain this and similar events, we propose a notch-triggered rotation mechanism whereby a relatively small subaerial calving event triggers a larger full-depth calving event due to the abrupt increase in buoyant load and the associated stresses generated at the ice-bed interface. We investigate the notch-triggered rotation mechanism by applying a geometric perturbation to the subaerial section of the calving front in a diagnostic flowline model of an idealised glacier snout, using the full-Stokes, finite element method code Elmer/Ice. Different sliding laws and water pressure boundary conditions are applied at the ice-bed interface. Water pressure has a big influence on the likelihood of calving, and stress concentrations large enough to open crevasses were generated in basal ice. Significantly, the location of stress concentrations produced calving events of approximately the size observed, providing support for future application of the notch-triggered rotation mechanism in ice-sheet models.

## 1 Introduction

Iceberg calving from marine-terminating glaciers is an important component of the mass balance of the Greenland Ice Sheet. Calving accounted for a third of total mass loss between 2009 and 2012 (Enderlin et al., 2014). Moreover, calving is an important control on the flow dynamics of tidewater glaciers, reducing the backstress in the glacier snout region and leading to flow acceleration and dynamic thinning (Thomas, 2004). The acceleration of Jakobshavn Isbræ (JI) by a factor of 4 since 1995, for example, is linked with its continued calving retreat following the disintegration of its ice shelf (Joughin et al., 2012).

Current models of iceberg calving fail to capture the full range of observed processes, and as such the parameterisations applied within ice sheet models are limited. van der Veen (1996) proposed the empirical height-above-flotation criterion, whereby the glacier calves to a point where its terminus is some fixed height above the flotation thickness. Although this method successfully reproduced advance and retreat behaviour for Columbia Glacier (Vieli et al., 2001) and Helheim Glacier



(Nick et al., 2009), a major shortcoming was the inability for ice to thin below the flotation thickness and form an ice tongue. A more physically based approach (Benn et al., 2007a, b) assumed that crevasses penetrating to the waterline penetrate through the full glacier thickness. This simple theory has been used in many recent modelling studies (e.g. Otero et al., 2010; Nick et al., 2013; Cook et al., 2014). However, since crevasse depths are calculated based on the equilibrium between

longitudinal stretching and ice overburden pressure (Nye, 1957), calving in these models arises only as a result of ice flow dynamics. The effect of localised processes such as melt-water undercutting (Luckman et al., 2015) or ice-cliff collapse (Bassis and Walker, 2012), or of superbuoyancy upon near-terminus stresses is not captured in these models.

Buoyant forces have been proposed as a driver of large calving events observed at major Greenland Ice Sheet marine-
terminating glaciers. Full-depth, rotating slab calving events observed at Helheim Glacier resulted from buoyant flexure of the glacier snout and the propagation of basal crevasses (e.g. Murray et al., 2015). Wagner et al. (2016) also showed that applying a buoyant force to an elastic beam model of a glacier resulted in large basal tensile stresses, which were further amplified by the emergence of a submarine protrusion of the calving face due to sea surface melting.

A multiple-iceberg calving event was observed at JI in August 2009 (Walter et al., 2012) that is not fully explained by existing calving models, but which we propose is tied to buoyant force changes over the course of the multi-stage calving event. In this observation, the collapse of a subaerial portion of the ice cliff was followed minutes later by a much larger, full-depth, rotating slab calving event across the same section of the front. We consider a mechanism to explain this event whereby a substantial portion of the snout becomes buoyant immediately following a small subaerial calving event, which
we term "notch-triggered rotation". In this mechanism, visualised in Fig. 1, the sudden increase in buoyant load causes the snout to lift and rotate. The resultant basal tensile stresses initiate basal crevassing, which rapidly propagates through the full glacier thickness. This mechanism is similar to the "footloose" mechanism investigated by Wagner et al. (2014) and earlier proposed by Scambos et al. (2005) for the breakup of tabular icebergs. However, in this study we consider the very short timescales arising from abrupt changes in the geometry, and further we analyse the viscous stresses originating at the ice-bed
interface rather than the elastic stresses resulting purely from bending.

Using a diagnostic numerical glacier model, we investigate whether notch-triggered rotation is a plausible calving mechanism. With sophisticated prognostic models, calving criteria can be tested by application to real glacier geometries (e.g. Nick et al., 2010; Krug et al., 2014) and the calving rate response to various environmental forcings can be quantified
(e.g. Cook et al., 2014; Todd et al., 2014). Simpler diagnostic models, however, provide insight into iceberg calving mechanisms by resolving the internal stresses under instantaneously imposed geometries (e.g. Hanson and Hooke 2000, 2003; O'Leary and Christofferson, 2013). Here we use a diagnostic model that is able to quantify changes in the stress field induced by geometrical perturbations of the ice front.



## 2 Model setup

In this study we use a two-dimensional diagnostic flowline model of an idealised glacier snout, to determine whether the magnitude of stresses generated by the notch-triggered rotation mechanism is sufficient to result in calving. The mechanisms of crevasse propagation through the ice thickness are not examined. We apply the buoyant forcing in the model by cutting a

notch into the subaerial ice cliff to a length $l_n$ from the waterline to the surface (Fig. 2). The ice flow solution is calculated using the open source, full-Stokes, finite element Elmer/Ice modelling software (Gagliardini et al., 2013).

### 2.1 Ice flow model

Elmer/Ice calculates velocity and stress profiles within the glacier by solving the Stokes equations for an incompressible fluid:

$$\nabla \cdot \boldsymbol{u} = 0 \tag{1}$$

$$\nabla \cdot \boldsymbol{\tau} - \nabla \boldsymbol{p} + \rho_i \boldsymbol{g} = 0 \tag{2}$$

where $\boldsymbol{u}$ is the velocity vector, $\boldsymbol{\tau}$ the deviatoric stress tensor, $\boldsymbol{p}$ the pressure, $\rho_i = 918$ kg m$^{-3}$ the ice density and $\boldsymbol{g} = (0,0,-9.81)$ m s$^{-2}$ the acceleration due to gravity. The ice rheology is described using Glen's flow law which relates deviatoric stress to the strain rate ($\dot{\varepsilon}_{ij}$):

$$\tau_{ij} = 2\mu\dot{\varepsilon}_{ij} \tag{3}$$

The effective viscosity $\mu$ is defined as

$$\mu = \frac{1}{2}A^{-1/n}\dot{\varepsilon}_e^{(1-n)/n} \tag{4}$$

where $\dot{\varepsilon}_e^2$ is the square of the second invariant of the strain rate tensor and $n = 3$ is the commonly used exponent in Glen's

flow law. The Arrhenius factor $A$ is expressed as

$$A = A_0 \exp(-Q/RT') \tag{5}$$

where $A_0$ is a constant, $Q$ the creep activation energy, $R$ the universal gas constant and $T'$ the temperature of ice relative to the pressure melting point (Cuffey and Patterson, 2010). For symbols and values used in this study, see Table 1. The temperature of glacier ice is set at a constant -9°C.

### 2.2 Boundary conditions

We use typical boundary conditions for a tidewater glacier. Along the upper surface, as well as the rear and lower surfaces delineating the notch when one is present, or the subaerial portion of the ice front otherwise, we ignore atmospheric pressure and apply a stress-free boundary condition:

$$\sigma_{nn} = 0 \tag{6}$$

$$\sigma_{nt} = 0$$




where $\sigma$ is the Cauchy stress and subscripts $n$ and $t$ refer to normal and tangential directions. Hydrostatic pressure is applied at the ice front below the waterline:

$$p_w = -\rho_w g z \tag{7}$$

where $p_w$ is the water pressure, $\rho_w = 1028$ kg m$^{-3}$ the ice density and the vertical $z$ axis is centred at the waterline. At the rear boundary 10 km upstream, lithostatic pressure is applied along with an inflow velocity of 5000 m a$^{-1}$, chosen to roughly

5    match the flow speed of JI at a similar distance from the calving front (Vieli and Nick, 2011).

At the basal boundary, a choice of sliding laws was available for grounded ice. Weertman-type power laws are common in glacier modelling applications (e.g. Krug et. al., 2014; Cornford et. al., 2015). This law takes the form

$$\tau_b = C|u|^{m-1}u \tag{8}$$

with $\tau_b$ the basal drag, $C$ the Weertman friction coefficient and sliding exponent $m = 1/3$. Values of $C$ range from $10^5$ to $10^8$

10   Pa m$^{-1/3}$ s$^{1/3}$, which includes the more realistic range of modelled values of ~$10^6$ to ~$10^7$ Pa m$^{-1/3}$ s$^{1/3}$ determined from surface velocity observations around Greenland outlet glaciers (Lee et al., 2015).

Alternatively, a Coulomb-limited sliding law (Schoof, 2005; Gagliardini et al., 2007) can be applied (referred to as the "Schoof law" from here on in). This law accounts for the effect of water pressure through an effective pressure term $N = $

$-\sigma_{nn} - P_w$. Basal drag is expressed as

$$\tau_b = C_c \cdot N \left( \frac{\chi}{1 + \alpha\chi^q} \right)^{1/n} \tag{9}$$

where

$$\chi = \frac{u}{C_c{}^n N^n A_s} \tag{10}$$

and

$$\alpha = \frac{(q-1)^{q-1}}{q^q}. \tag{11}$$

$C_c = 1$ is the maximum value of $\tau_b/N$, $q = 1$ is the post-peak exponent, $A_s$ is the value of the sliding coefficient in the absence of cavitation and $n$ is the flow law exponent. As in previous studies (e.g. Nick et al., 2010; Krug et al., 2014) a free

hydrological connection is assumed between the subglacial drainage system and the sea, so hydrostatic water pressure is applied at the ice-bed interface.

The contact problem (Durand et al., 2009) is solved at the ice-bed interface to determine where ice is grounded or floating. In this implementation, nodes touching the bedrock where the normal stress exerted by the ice is greater than the seawater

pressure ($\sigma_{nn} > p_w(z_b)$) are considered grounded and have zero vertical velocity, while nodes that have separated from the bedrock or where $\sigma_{nn} \leq p_w(z_b)$ are floating and can have non-zero vertical velocity. The model is initialised with the glacier fully grounded along its entire length.

## 3 Model Results

Experiments were run for a glacier with water depth $d_w = 900$ m, terminus thickness $h_t = 980$ m, surface slope $\alpha = 3°$ and $C = 2.371 \times 10^6$ Pa m$^{-1/3}$ s$^{1/3}$, with notches cut to varying lengths. For these experiments, full hydrostatic pressure (Eq. 7) was applied along the basal boundary. Figure 4 shows longitudinal deviatoric stresses $\tau_{xx}$ mapped for the $l_n = 100$ m case, with the $l_n = 0$ m case mapped in Fig. 3 for comparison. Basal stresses are plotted for $l_n = 0$ m, $l_n = 80$ m and $l_n = 100$ m (Fig. 5). An estimate of the tensile strength of glacier ice from Vaughan (1993) is also plotted to provide estimates of the critical stress required for basal crevasse opening.

Notch cutting caused basal ice to become ungrounded between 191 m and 644 m upstream of the ice front for $l_n = 100$ m (Fig. 4). Prominent stress concentrations associated with ungrounding and regrounding also appeared at the basal boundary which were not present in the unperturbed case. These stresses exceed the tensile strength of ice and would therefore result in crevasse initiation. The tensile stress peak centred around 644 m upstream of the ice front resulted from separation of basal ice from the bedrock, as the buoyant snout tended to lift. The abrupt change in basal drag across the grounding zone, where ice that has separated from the bedrock accelerates, gave rise to this stress peak. Further notch-cutting caused this stress peak to shift upstream and increase in magnitude.

The region of compressive stress centred around 191 m upstream of the ice front arose from ice regrounding on the bedrock downstream of the grounding line, due to the backstress applied on the ice front by hydrostatic pressure. An imbalance between the hydrostatic and cryostatic pressure normal to the terminus tends to warp the snout downwards (see Fig. S1; Reeh, 1968), with the same effect seen at the start of prognostic model runs by Benn et al. (2017). Experiments in which the hydrostatic pressure from the pro-glacial water body was removed, or the bedrock lowered downstream of the grounding line, did not include this compressive stress peak while still featuring the tensile stress peak, supporting our assertion that the compressive stress concentration resulted from basal ice regrounding.

Corresponding longitudinal velocity maps for the frontal region are shown in Fig. S2 (unperturbed) and Fig. S3 ($l_n = 100$ m). There is an acceleration of ~2000 m a$^{-1}$ following the notch cutting, resulting from the reduced basal friction in the ungrounded region.

A critical notch length $l_{crit}$ was required before the glacier snout became buoyant and the tensile stress peak appeared. A sensitivity study was carried out to explore the relationship of this critical notch length to the bed stickiness and the glacier surface slope (Fig. 6), and to determine how this relationship affects the maximum basal stress (Fig. 7). Setting the notch length $l_n = l_{crit}$ resulted in a noisy maximum basal stress signal so we instead set $l_n = l_{crit} + 25$ m which allows a coherent pattern to emerge.





Ungrounding occurred even without a notch on glaciers with very slippery beds for all surface slopes, and at all values of the friction coefficient for a 2° surface slope. For steeper surface slopes the critical notch length increased with bed stickiness before levelling off. For a given value of the friction coefficient, the critical notch length also increased with surface slope. Similarly, the maximum basal stress increased with both friction coefficient and surface slope. For very slippery beds the maximum stress was below the upper boundary of the tensile strength envelope, but significantly it was above the critical stress for crevasse initiation through the realistic range of friction coefficients $C = 10^6$ to $10^7$ Pa m$^{-1/3}$ s$^{1/3}$.

These experiments reveal a complex picture of the conditions that favour calving. An explanation for the relationship between the critical notch length and bed stickiness does not readily present itself, and this effect may warrant further investigation. The relationships of surface slope with both the critical notch length and the maximum basal stress are more easily explained. The terminus of a steeper sloped glacier is more strongly grounded, requiring the removal of more ice to render it buoyant, than a more gently sloping glacier. The longer submarine foot and larger buoyant forces that result then favour larger basal stresses (e.g. Wagner et al., 2016).

## 3.1 Water pressure dependency

Tidewater glaciers such as JI are subject to the influence of water pressure where they meet the ocean, therefore it is appropriate to examine the region around the grounding line and calving front using a water pressure dependent sliding law. To investigate the effects of water pressure upon the notch-triggered rotation mechanism, experiments were conducted using the Schoof law (Eq. 9). In all following experiments a similar setup as before was used, with $\alpha = 3°$, a varying notch length, and the sliding coefficient $A_s = 3.169 \times 10^{-21}$ Pa$^{-3}$ m s$^{-1}$. Experiments F0 and F100 were carried out with full hydraulic connectivity at the ice-bed interface, and experiments Z0 and Z100 with zero hydraulic connectivity (i.e. $p_w = 0$ everywhere). See Table 2 for details of parameters used in Schoof law experiments.

The resulting stress profiles for these experiments (Fig. 8) are highly dependent on the basal water pressure, with experiments Z0 and Z100 exhibiting stress patterns identical to the Weertman law experiments. However with full water pressure applied (F0 and F100), there is a region of large tensile stress that exists independent of any perturbation. Notch cutting has minimal impact on the magnitude or location of this region. This region of large stress exists because the basal shear stress in the frontal region is small, since the effective pressure is zero; therefore, the basal shear stress is increased upstream, and this upstream transferal of stress occurs via a region of increased englacial tensile stress. The magnitude of these stresses suggests an inherent instability for glaciers in such a configuration when subject to full basal water pressure.

The assumption of perfect hydraulic connectivity, however, may not hold for large distances upstream of the grounding line (Cuffey and Paterson, 2010, p. 283). We therefore carried out additional experiments P0 and P100 to simulate limited





hydraulic connectivity by linearly reducing the water pressure at the ice-bed interface from full hydrostatic pressure at the front to zero at the rear of the domain (Table 2), similarly to Leguy et al. (2014). Experiment P0 shows a region of large tensile stress, like those seen in experiments F0 and F100 (Fig. 9). The notch perturbation in experiment P100 results in a Weertman-like stress peak which is significantly larger in magnitude than the unperturbed stress peak.

5 **4 Discussion**

Our experiments show that perturbations to the ice front geometry can induce large stress concentrations in basal ice. The magnitude and location of these stress concentrations shows a strong dependency on the basal drag. For a glacier snout already close to flotation, only a relatively small perturbation was required to induce large stresses. This is in line with the observed relationship between calving rate and water depth (Brown et al., 1982).

The choice of diagnostic model for a calving study was criticised by Cook et al. (2014) after their prognostic model showed much greater sensitivity to atmospheric as opposed to oceanic forcing than diagnostic models (O'Leary and Christoffersen, 2013), suggesting that this was due to the inability of a diagnostic model to respond to stress perturbations through ice deformation. However, over the short timescales of interest in this study, deformation of ice is negligible. In our
experiments, measured vertical velocities for the ungrounded regions of basal ice were of the order ~10 m a$^{-1}$, equating to ~0.1 mm of lifting over 5 minutes, which would have negligible effect on the stress field. Therefore, our choice of a diagnostic model is an appropriate one for this study.

As in other diagnostic studies we did not apply a calving criterion, instead using the location of basal stress peaks as an
20 indication of where crevasses may form. For this to result in calving on the timescale proposed requires the assumption of full-thickness crevassing on timescales much faster than those observed by e.g. Murray et al. (2015). Given a sufficiently large buoyancy force, this assumption can be held as true, as once a crack has initiated, the tensile stress which opened that crack refocuses at the crack tip causing it to continue to propagate. As the crevasse increases in height, hydrostatic pressure acting to open the crevasse decreases at a faster rate than the ice overburden pressure acting to close it; therefore, larger basal
stresses are required for full-depth crevassing than for crevasse initiation. However, once a crevasse has started to propagate and the downstream portion of the snout has begun to lift and rotate, elastic stresses further contribute to the crevasse growth in a feedback process. Benn et al. (2017) reported that glacier geometries that did not result in calving in Elmer/Ice via crevasse depth calving laws still produced large full-depth calving event when exported into HiDEM, a model representing glacier ice as a lattice of particles connected by breakable elastic beams. Further investigation of the rate and modes of
crevasse propagation could integrate Linear Elastic Fracture Mechanics into a glacier model featuring basal crevasses (van der Veen, 1998), or use a model such as HiDEM in conjunction with Elmer/Ice (Benn et al., 2017).



Our study builds on that of O'Leary and Christoffersen (2013), which also explored the effect of geometrical perturbations at the ice front on the likelihood of calving. Their study found that undercutting led to larger calving events and a higher overall calving rate, which appears to be at odds with the results presented here: undercutting would reduce the buoyant load and potentially stabilise the terminus. Our results differ because the sharp transition in basal drag is not possible at the stress-free surface boundary. Furthermore, the geometry of their model was set up to explore surface crevassing while ours was designed to explore basal crevassing. In reality a mixture of these effects may be working together to promote or prohibit calving.

Figure 1 suggested the subaerial calving event may result from undercutting by a waterline notch. Although this process is observed at some glaciers (e.g. Kirkbride and Warren, 1997; Röhl, 2006) it is questionable whether it could be a major factor in the Ilulissat Icefjord (where the original observation was made), in which the loosely bonded mélange in summer may act to damp any wavecutting action (Amundson et al., 2010). An alternative potential mechanism for triggering the subaerial calving event is provided by spontaneous collapse of the ice cliff. The maximum stable cliff height for damaged glacier ice was calculated by Bassis and Walker (2012) as 110 m while Hanson and Hooke (2003) suggested a maximum stable height of ~ 70 m based on diagnostic model experiments. The ice cliffs of JI approach 100 m but rarely exceed this height, suggesting that the inherent instability of ice cliffs may be the limiting factor and could trigger calving through notch-triggered rotation.

Buoyancy in a glacier snout can also be induced by thinning due to high surface melt rates. However, the almost immediate increase of buoyant load resulting from the subaerial calving event proposed here occurs on timescales much faster than can be accommodated by ice creep, leading to a higher probability of calving (e.g. Boyce et al., 2007). The specific location of the basal stress peak varied with many parameters including the notch length but tended to be within one ice thickness of the terminus, consistent with observations (e.g. Walter et al., 2012, Murray et al., 2015). The location of the peak stress always occurred much further back from the terminus than the cliff at the rear of the notch, leading to an amplification of the original subaerial calving event. The value of this amplifying factor cannot be accurately quantified within the limitations of a diagnostic model; however, it may present a method of linking environmental forcings to the calving rate.

There are a number of possible refinements to our model. We ignored lateral drag, which combines with basal drag to support the driving stress. Although lateral drag may be negligible along the flowline of wide ice streams, JI was able to form a floating tongue so it must be assumed that lateral drag is significant (e.g. Thomas, 2004). Its omission may have caused the model to overstate the dependence of basal stresses on the basal sliding law. Our model also omits the effect of temperature. The viscosity of ice and transmission of stresses are dependent on thermal gradients. JI has large vertical temperature gradients (Lüthi et al., 2002) and temperate basal ice, which are thought to play a role in its fast sliding. Warmer basal temperatures may act to damp the intensity of basal stress concentrations.





Notch-triggered rotation mechanism was shown to be irrelevant under the full Schoof regime, since a glacier in these conditions would tend to be vulnerable to buoyant calving anyway. This raises the question of whether the Schoof law with full water pressure provides an accurate representation of basal sliding for JI. We expect low effective pressure in the frontal region, however given that the glacier snout is mostly grounded in the summer (e.g. Amundson et al, 2010), perfect hydraulic connectivity cannot be assumed along the ice-bed interface. Complete suppression of water pressure at the ice-bed interface resulted in a basal stress pattern identical to the Weertman case. With $C_c = q = 1$ and $m = 1/n$ as in this study, it can be easily shown that large $N$ (~10 MPa in the absence of water pressure) leads to small $\chi$ and Eq. (9) reduces to a Weertman power law:

$$\tau_b = \left(u_b A_s^{-1}\right)^{1/n}. \tag{12}$$

On the other hand, limiting the basal water pressure without supressing it completely (experiments P0 and P100) resulted in a transition case displaying behaviour from both the Weertman and Schoof regimes; the unperturbed stress profile was similar to the Schoof case, but the perturbation resulted in a significantly larger Weertman-like stress peak. This raises the possibility that a lightly grounded glacier snout, already in a state of basal tension, could be subjected to high enough stress by a minor subaerial calving event, like that observed at JI (Walter et al., 2012), to cause full depth crevassing and buoyant calving.

**5 Conclusions**

Our results show that the notch-triggered rotation mechanism does produce calving for an idealized marine-terminating glacier. Although notch-triggered rotation did not significantly affect stresses when applying the Schoof law under full hydrostatic pressure, removing the assumption of perfect hydraulic connectivity at the ice-bed interface significantly enhanced the likelihood of calving through this mechanism. Significantly, a realistic length scale for calving events, on the order of hundreds of meters and generally less than one ice thickness, naturally results from the model physics. Fast flowing glaciers near flotation and with shallow surface slopes may be especially vulnerable to buoyant calving due to basal crevassing. The notch-triggered rotation mechanism proposed here to explain the observed calving event (Walter et al., 2012) does not replace other models of calving. Instead, it bolsters our understanding of calving by providing insight into multi-stage calving events occurring particularly on large, fast-flowing tidewater glaciers.

**Author contribution**

M.T. designed the experiments with contribution from A.J.P. and S.L.C and M.T. carried them out. M.T. prepared the manuscript with contributions from all co-authors.



**Competing interests**

The authors declare that they have no conflict of interest.

**Acknowledgements**

M.T. was supported by the Natural Environment Research Council (NERC) through the Great Western Four+ (GW4+)
Doctoral Training Partnership. We thank Olivier Gagliardini and Thomas Zwinger for running the free Elmer/Ice training
course at the Neils Bohr Institute, Copenhagen in November 2015. We thank Olivier Gagliardini in particular for suggestions
on the study scope and invaluable technical support. We also thank Till Wagner for useful comments on the original
manuscript.

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

30





| Parameter | Symbol | Value | Units |
|---|---|---|---|
| Arrhenius factor | $A$ | | Pa$^{-3}$ s$^{-1}$ |
| Coulomb sliding coefficient | $A_s$ | $3.169 \times 10^{-21}$ | Pa$^{-3}$ m s$^{-1}$ |
| Arrhenius prefactor | $A_0$ | 1916 | Pa$^{-3}$ s$^{-1}$ |
| Weertman friction coefficient | $C$ | $10^5 - 10^8$ | Pa m$^{-1/3}$ s$^{-1/3}$ |
| Maximum value of $\tau_b/N$ | $C_c$ | 1 | |
| Water depth | $d_w$ | 900 | m |
| Acceleration due to gravity | $g$ | 9.81 | m s$^{-2}$ |
| Terminus thickness | $h_t$ | 980 | m |
| Critical notch length | $l_{crit}$ | | m |
| Notch length | $l_n$ | | m |
| Weertman sliding exponent | $m$ | 1/3 | |
| Effective pressure | $N$ | | Pa |
| Glen's flow law exponent | $n$ | 3 | |
| Pressure tensor | $\mathbf{p}$ | | Pa |
| Water pressure | $p_w$ | | Pa |
| Post-peak exponent | $q$ | 1 | |
| Creep activation energy | $Q$ | 139 | kJ mol$^{-1}$ |
| Universal gas constant | $R$ | 8.314 | J K$^{-1}$ mol$^{-1}$ |
| Pressure-adjusted temperature | $T'$ | | K |
| Velocity tensor | $\mathbf{u}$ | | m s$^{-1}$ |
| Glacier surface gradient | $\alpha$ | $2 - 5$ | ° |
| Strain rate tensor | $\dot{\boldsymbol{\varepsilon}}$ | | s$^{-1}$ |
| Square of 2$^{nd}$ invariant of $\dot{\boldsymbol{\varepsilon}}$ | $\dot{\varepsilon}_e^2$ | | s$^{-2}$ |
| Effective viscosity | $\mu$ | | Pa s |
| Ice density | $\rho_i$ | 918 | kg m$^{-3}$ |
| Water density | $\rho_w$ | 1028 | kg m$^{-3}$ |

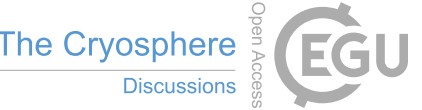



| | | |
|---|---|---|
| Cauchy stress tensor | $\boldsymbol{\sigma}$ | Pa |
| Deviatoric stress tensor | $\boldsymbol{\tau}$ | Pa |
| Basal drag | $\tau_b$ | Pa |

**Table 1.** Symbols and values of physical and numerical constants and parameters used in this study.



| Experiment | Hydraulic connectivity | $l_n$ (m) | $x_1$ (m) | $x_0$ (m) |
|---|---|---|---|---|
| F0 | Full | 0 | 10000 | 10000 |
| F100 | Full | 100 | 10000 | 10000 |
| Z0 | Zero | 0 | 0 | 0 |
| Z100 | Zero | 100 | 0 | 0 |
| P0 | Partial | 0 | 0 | 10000 |
| P100 | Partial | 100 | 0 | 10000 |

**Table 2. Hydraulic connectivity along the ice-bed interface for experiments using the Schoof law. Water pressure is 100% of the full hydrostatic pressure (Eq. 7) downstream of position $x_1$. Between $x_1$ and $x_0$ water pressure reduces linearly to 0%.**

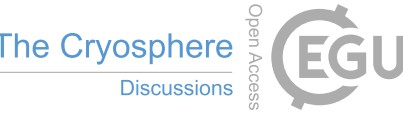



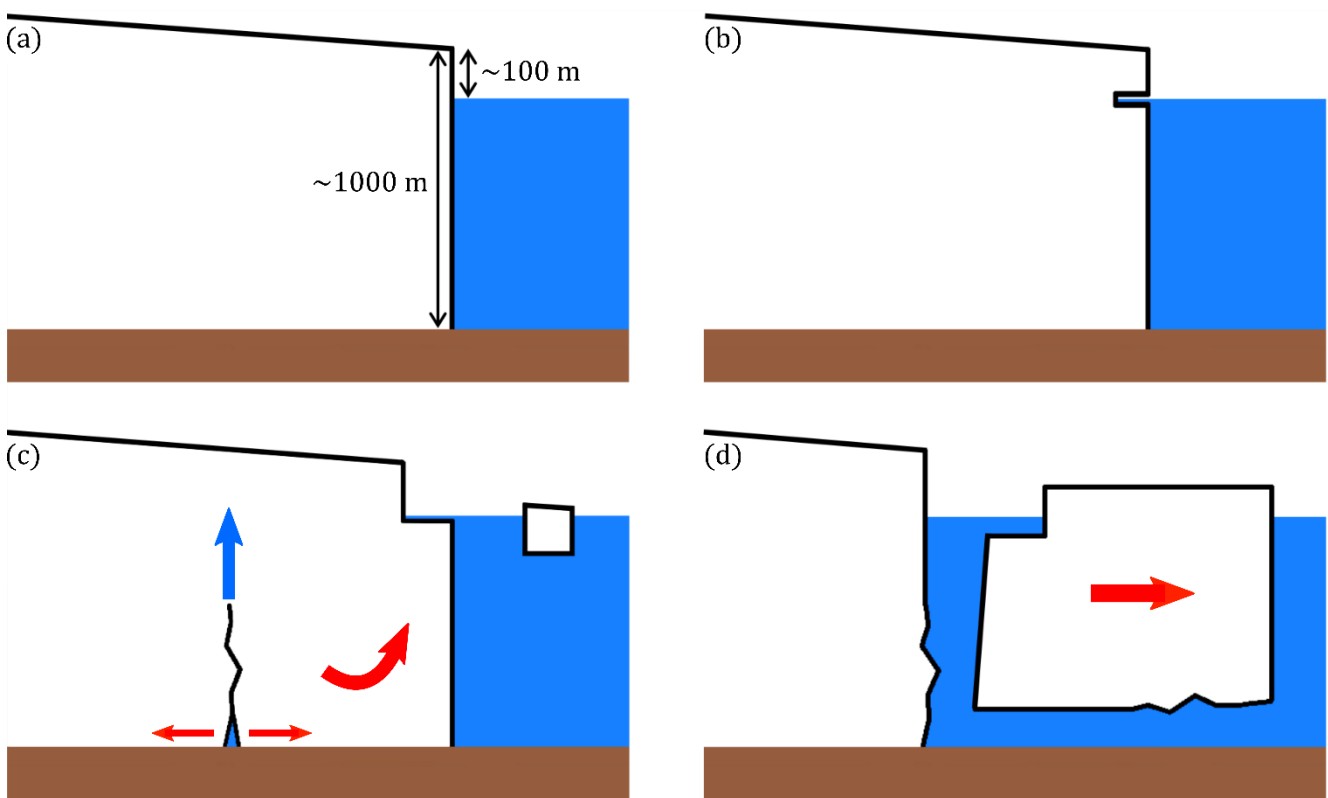

**Figure 1. Proposed calving mechanism. (a) Lightly grounded terminus of a tidewater glacier with approximate dimensions of e.g. Jakobshavn Isbræ. (b) A weakness develops in the subaerial section of the front due to (e.g.) undercutting by a wave-cut notch at the waterline. (c) A small subaerial calving event rapidly increases the buoyant load, causing the terminus to tend to lift and rotate. Basal crevasses open and propagate rapidly upwards. (d) Full-depth crevassing results in a large, rotating-slab calving event. The long-term calving rate is driven by the notch melt rate but is amplified by an unconstrained factor.**





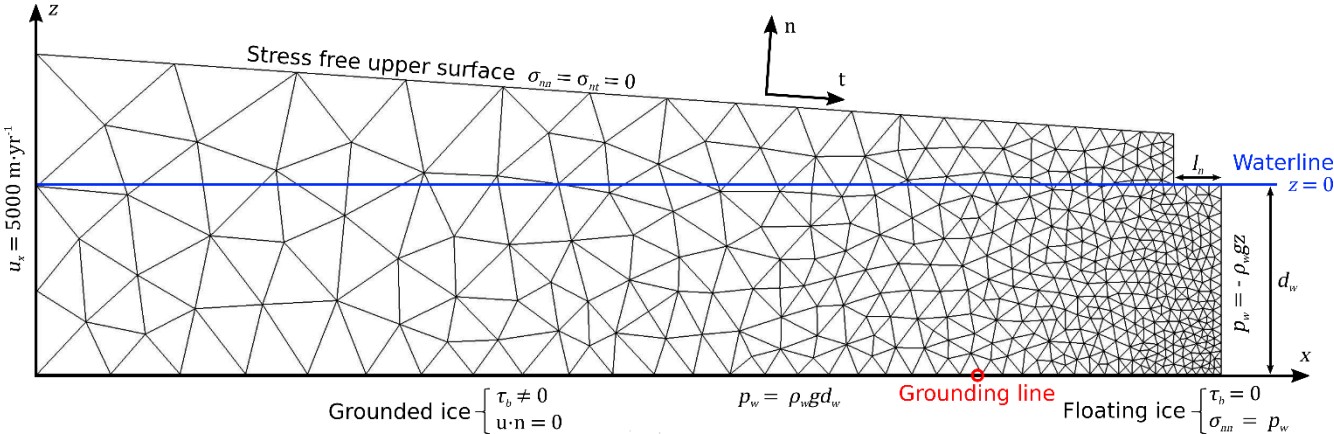

**Figure 2. Example mesh and boundary conditions (not to scale). Mesh resolution increases close to the calving front and basal boundaries. Symbols: normal stress $\sigma_{nn}$, shear stress $\sigma_{nt}$.**





**Figure 3. Longitudinal deviatoric stress map of the terminus of the glacier before the cutting of a notch, with contours at 0.1 MPa spacing. Note the qualitative similarity to stress maps presented in Hanson and Hooke (2000, 2003). $d_w$ = 900 m, $h_t$ = 980 m, $\alpha$ = 3° and C = 5.623 x 10$^6$ Pa m$^{-1/3}$ s$^{1/3}$.**







**Figure 4. Longitudinal deviatoric stress map of the terminus of the glacier with a notch cut to a length $l_n$ = 100 m, with contours at 0.1 MPa spacing. $d_w$ = 900 m, $h_t$ = 980 m, $\alpha$ = 3° and C = 5.623 x 10⁶ Pa m⁻¹ᐟ³ s¹ᐟ³. Ungrounding occurred between 191 m and 644 m.**





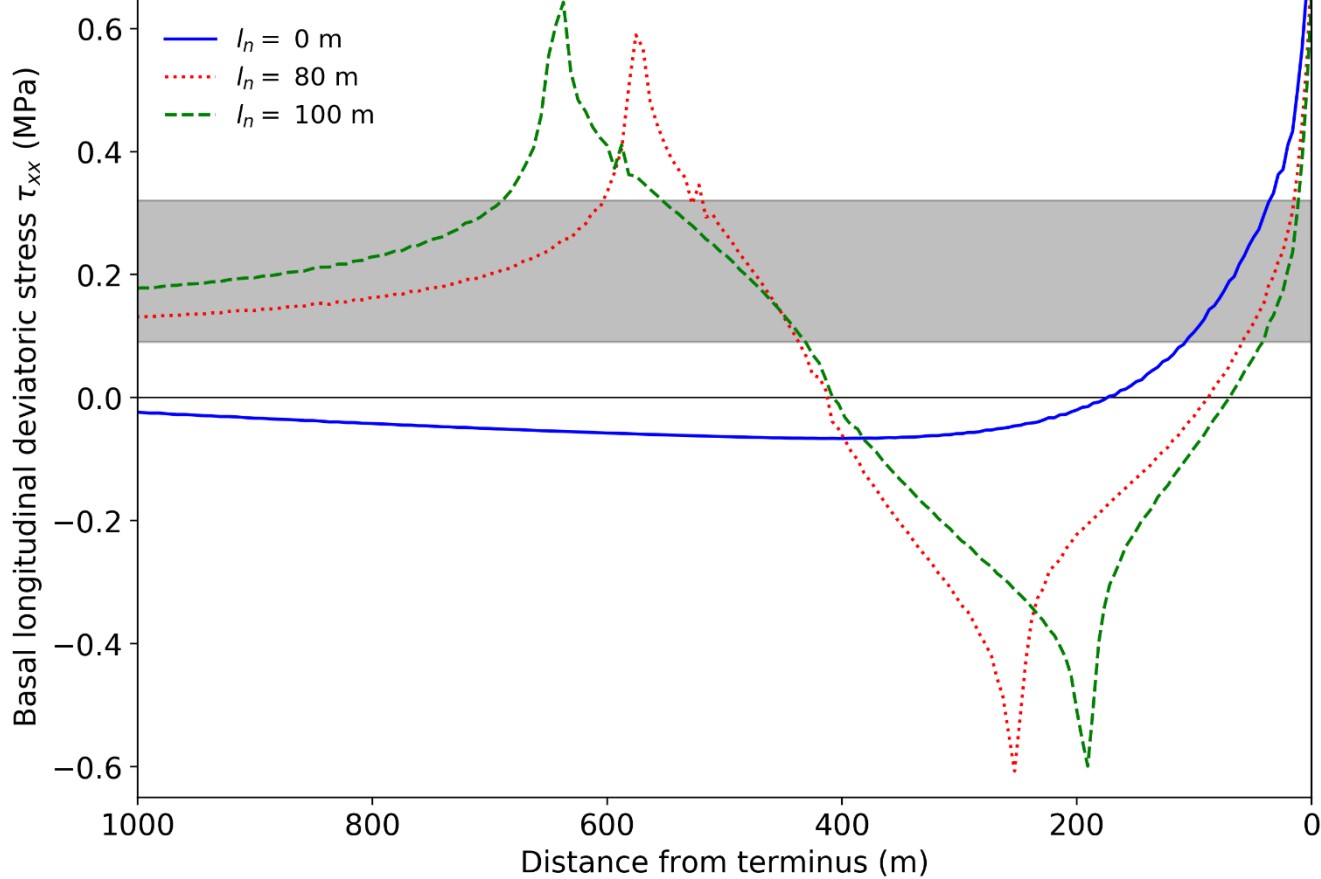

**Figure 5. Plots of basal longitudinal deviatoric stress for $l_n$ = 0 m, 80 m and 100 m. $d_w$ = 900 m, $h_t$ = 980 m, $\alpha$ = 3° and C = 5.623 x $10^6$ Pa m$^{-1/3}$ s$^{1/3}$. The shaded region denotes the tensile strength envelope calculated from Vaughan (1993). The large basal stress concentrations from Fig. 4 correspond to the peak and trough in the $l_n$ = 100 m plot. Ungrounding occurred between 253 m and 575 m for $l_n$ = 80 m, and between 191 m and 644 m for $l_n$ = 100 m. Note that for this setup, the critical notch length $l_{crit}$ = 79 m.**



**Figure 6. Stress switching notch length $l_{crit}$ plotted for a range of Weertman coefficients $C$ and surface gradients $\alpha$.**







**Figure 7. Basal stress maximum plotted with the notch length equal to $l_{\mathrm{crit}}$ + 25 m across a range of Weertman coefficients and surface gradients. The shaded region denotes the tensile strength envelope calculated from Vaughan (1993).**







**Figure 8.** Comparison of basal stresses using the Coulomb sliding law with full (red lines) and zero (blue lines) water pressure at the ice-bed interface, before (solid lines) and after (dashed lines) cutting of a 100 m notch. Ungrounding occurred between 200 m and 381 m for F0, 16 m and 510 m for F100, and between 243 m and 612 m for Z100. No ungrounding occurred for Z0.



**Figure 9.** Comparison of basal stresses using the Coulomb sliding law partial hydraulic connectivity at the ice-bed interface, before (solid blue line) and after (red dashed line) cutting of a 100m notch. Ungrounding occurred between 158 m and 637 m for P100 and not at all for P0.

