# Peer review of "Buoyant forces promote tidewater glacier iceberg calving through large basal stress concentrations"

_The Cryosphere, 2018_

## Referee Comment (RC1) · Anonymous Referee #1 · 1 Dec 2018

I'm sorry, but so much of the discussion in the manuscript depends on un-referenced and un-explained "references to previous work" that I think that the manuscript needs to be revised significantly. The specific places where the discussion and explanations are inadequate are listed below.

Other than the expository problem with the manuscript. I find the science compelling and well done. The work is creative and important in the study of iceberg calving mechanisms.

Here's the stuff that needs attention (in my estimation):

I wonder if the title really does justice... the paper is about bending moments (viscous and plastic bodies have bending moments too!) generated by geometry changes at the

ice front due to ice/ocean and ice/atmosphere and ice/wave interactions... the present title could be misunderstood to represent "same old basal shear stress" stuff....

line 13 - would it be more accurate to say "viscous bending moment" (remember you can bend a beam viscously and elastically and viscoelastically) leading to high tensile stress concentration at the bed... instead of stresses at the ice-bed interface? Who cares what the stresses are at the interface if the ice is actually in a state of bending induced fracture?

line 29 - Would this be a place to add a reference to Weertman?

line 18 page 2 - The rotation should be indicated as "bottom out"...

line 8 page 3 - If I were to be pedantic, I would say that a reference should be given for "Stokes equations" (in actuality, Stokes was prolific and probably has many equations associated with his name). Ditto for "Glen's flow law"... a reference should be given.

line 1 page 4 - Is Cauchy stress the same as deviatoric stress?

line 9 page 5 - Just out of curiosity why are 191 and 644 meters so precisely known as to be significant to the single meter? Can the authors tell us what would happen if the numbers were 192 and 643?

section 3.1 - What is the a priori reason to expect water pressure to be significantly important in the problem? is it for promoting fracture propagation or is it for lubricating the base?

line 4 page 7 - What is "Weertman-like"???? This seems to come in out of the blue... Weertman published hundreds of papers in his life, what is referred to here?

line 1 page 9 - Notch-triggered rotation mechanism was shown to be irrelevant under the full School regime"... Readers will get confused here, because "full School regime" is a weak form of jargon that doesn't really convey the precise ideas (regardless of whether they are published in Schoof... my hunch is that the authors have a different

meaning, i.e., an interpretation that they ascribe to the term "full School regime")

line 12 and 13 page 9 - "Weertman . . . regime"???? Weertman-like stress peak???? What is this, and why the name Weertman???? Seems like citations and explanations are required. The discussion is flawed because it relies on readers having prior knowledge of what a "Weertman regime" is...

I don't really see the necessity for me to review the manuscript again if the above items are handled by the authors in a revision.

---

## Referee Comment (RC2) · Bassis (Referee) · 16 Jan 2019

Bassis (Referee)

jbassis@umich.edu

This manuscript describes a set of flowline "full Stokes" simulations of notch induced buoyant calving with geometries analogous to Jakobshavn Isbrae, one of the largest outlet glaciers draining the Greenland Ice Sheet. The authors hypothesize that buoyant forces associated with the detachment of a block of ice above the waterline could trigger a buoyant calving event, similar to a multi-part calving event observed.

The idea behind buoyant calving events being triggered by subaerial block detachment is old and goes back (at least) as far as the early workshops on calving. Where this manuscript goes beyond previous research is in not only quantifying the stresses associated with block detachment using a model, but also probing the relationship of the

near terminus stress field with the choice of sliding law. Moreover, the authors attempt to further link the state-of-stress with variations in sub-glacial water pressure, which has been under explored to date.

In general I found the manuscript straightforward and easy to read. There are a couple of issues, however, that I do want to point out in addition to a few terminology suggestions. Most of them are relatively minor and should be easy to correct.

Major suggestions:

There is one major issue with the analysis, which is that the authors compare longitudinal deviatoric stress with a yield strength estimate from Vaughan (1993). The longitudinal deviatoric stress is problematic for two reasons. First, the various components of the deviatoric stress are not coordinate system invariant and have little physical meaning: a different coordinate system would result in different numbers. It is possible that the authors want to look at, say the components of the traction along the bed (which is well defined) or the largest principle deviatoric stress (which is also well defined). But this raises the more fundamental issue: it is the largest principle *Cauchy stress* and not the deviatoric stress that controls *tensile* fracture. And it is clear from the manuscript that the authors are fully focused on tensile basal crevasses. If the authors want to argue that the stresses are sufficient to trigger a *tensile* basal crevasse then they need to examine the largest principle *Cauchy* stress. Fortunately, this should be straightforward to compute from the full Stokes model. More problematically for the analysis performed here, for a kilometer thick glacier, the hydrostatic pressure is probably of the order of 10 MPa and may result in a negative (compressive) largest principle Cauchy stress. I should point that this is a common problem when dealing with failure of ice and especially basal crevasses. The most common solution to this problem is to (rather arbitrarily) superpose a hydrostatic pressure associated with water to the largest principle Cauchy stress to simulate the effect of water filled crevasses. This is commonly done and I think the authors could get away with it here if they want. Technically, you can't really do this and the right way to do it is to calculate the Cauchy stress after

introducing an infinitely narrow test crack. Doing it the right way, usually results in a compressive stress when using the power-law creep rheology of ice. If the authors go the usual route of superposing a hydrostatic stress field, I do suggest showing the stress with and without water pressure to emphasize that the water pressure is (or is not) critical.

Another aspect of the analysis that is somewhat problematic is that the authors are comparing their stress metric to the yield strength estimated by Vaughan (1993). My understanding, however, is that Vaughan (1993) examined various yield strength envelopes, finding that the Von Mises stress envelope provided the best fit to the observations. The Von Mises yield criterion, however, is only equivalent to tensile failure in uniaxial loading, which is not the case for the model considered here. Recalling that the second deviatoric stress invariant invariant is proportional to the Von Mises stress, what I suggest is that in addition to the largest principle Cauchy stress, the authors also consider showing the second deviatoric stress invariant as an additional stress metric. This stress metric can be more directly compared with Vaughan's estimated yield strength. Note that in two dimensions, the second effective deviatoric stress invariant is equal to the maximum shear stress and thus the failure mechanism predicted by this envelope would be shear, rather than purely tensile failure and, if the authors go this route, the authors will need to be careful to point this out. Although we speculated that shear failure is important for tall calving cliff in Bassis and Walker (2012), I'm not aware of any strong observational evidence supporting shear failure in calving so the authors may want to take this suggestion under advisement as the broader community has doubts about the viability of shear failure.

Another minor point is that the authors convincingly argue that time scales they are interested are short compared to the time scales of flow and thus they can ignore the effect of ice flow on their experiments. However, if the time scale of interest is short compared to the time scale of flow, then this would seem to imply that an elastic rheology would be appropriate. This is surprising to most, but the elastic stress can

be quite different from the viscous stress and this is primarily a consequence of the non-linearity in the creep flow law used.

Minor comments. Page 3, near line 20: Technically, you can impose a boundary condition on the traction and not on the individual stress components.

What is a full School regime?

The authors should be a little bit careful when discussing sliding laws, water pressure and the stress regime because glaciers are actually three-dimensional with bumps in the bed. In three-dimensions, these bumps play a pretty big role in controlling the stress transmission upstream because portions of the calving front maybe well grounded whilst other portions are close to flotation.

---

## Author Comment (AC1) · 13 Feb 2019

**Author's response to referee comments on "Buoyant forces promote tidewater glacier iceberg calving through large basal stress concentrations"**

***Authors*: Trevers, M., Payne, A. J., Cornford, S. L., Moon, T.**

The Cryosphere Discuss., https://doi.org/10.5194/tc-2018-212

We would like to thank both reviewers for their constructive and insightful criticisms and comments on our manuscript. These will be taken account of in our revised manuscript. Here we reply to the more important criticisms.

As both reviewers correctly point out, there are numerous instances where we have made use of jargon phrases which have been poorly explained and are therefore confusing to the reader. These will be more clearly defined to improve clarity for the reader. Typos will also be addressed, and appropriate references for Weertman sliding, Stokes equations and Glen's Flow Law will also be included where necessary.

Below, we deal with substantive comments from both reviews. Reviewer comments, numbered chronologically, are in italics, and the author response in plain text. Details of changes to the manuscript to address comments will be highlighted in bold text.

**Reviewer #1**

1. *I wonder if the title really does justice. . . the paper is about bending moments (viscous and plastic bodies have bending moments too!) generated by geometry changes at the ice front due to ice/ocean and ice/atmosphere and ice/wave interactions. . . the present title could be misunderstood to represent "same old basal shear stress" stuff. . ..*

The paper is not specifically about bending moments but rather about forces generated at the bed instantaneously at the onset of bending in response to the perturbation. Some further explanation is required on this for clarity (see the response to the next comment), but as such, we do not intend to change the title.

2. *line 13 - would it be more accurate to say "viscous bending moment" (remember you can bend a beam viscously and elastically and viscoelastically) leading to high tensile stress concentration at the bed. . . instead of stresses at the ice-bed interface? Who cares what the stresses are at the interface if the ice is actually in a state of bending induced fracture?*

As above, we don't attribute these stresses to a bending moment. We attribute these longitudinal stresses as those required to balance the abrupt decrease in basal shear across the grounding zone. Arguments to support this are as follows.

Firstly, the region of large stress is very sharply focused at the bed. In the bending moment hypothesis, they would be further distributed into the ice body, with a more gradual increase towards the bed. Secondly, the location of the maximum stress corresponds with the precise point of ungrounding. In the bending hypothesis, it would be slightly downstream of this point due to a finite radius of curvature. Thirdly, the downstream longitudinal deviatoric stress associated with regrounding of glacier ice is negative (corresponding to the abrupt increase in basal shear stress at the point, the inverse of the effect occurring upstream), and not positive as it would be under the bending hypothesis. Finally, we show that the form and magnitude of stress is dependent upon the choice of sliding law and application of basal water pressure, which would be largely irrelevant under the bending hypothesis.

**We will include these arguments in the discussion to make our interpretation clear.**

3. *line 1 page 4 - Is Cauchy stress the same as deviatoric stress?*

The deviatoric stress indicates the deviation from the Cauchy (or full) stress, i.e. it negates the average pressure term. For our ~1km thick glacier, the pressure will be ~10MPa everywhere along the bed and therefore the Cauchy stress, ignoring additional water pressure, will be compressive everywhere.

**We will clarify this detail in the paper with an adjustment to equation 2.** However, the stress metric we will be using in the final revision of the paper will no longer the longitudinal deviatoric stress but rather the largest principle Cauchy stress, in response to comments by Reviewer #2.

4. *line 9 page 5 - Just out of curiosity why are 191 and 644 meters so precisely known as to be significant to the single meter? Can the authors tell us what would happen if the numbers were 192 and 643?*

The precision of these numbers is not important to our argument, and is in a small part determined by the mesh resolution (i.e. the peak occurs at a mesh node). Since it is not possible to identify locations to the nearest meter in the figures provided, **we will change to reporting them as "approximately 190 m" and similarly for other such numbers.**

5. *section 3.1 - What is the a priori reason to expect water pressure to be significantly important in the problem? is it for promoting fracture propagation or is it for lubricating the base?*

The significance is that the mechanism we suggest relies on the abrupt reduction in basal shear stress where the ice ungrounds to produce the longitudinal stresses to balance this. If we apply a sliding law where the basal shear stress reduces gradually as a function of effective pressure, we wouldn't expect the same result. **We will add a sentence to the manuscript to explicitly make this point.**

**Reviewer #2**

1. *There is one major issue with the analysis, which is that the authors compare longitudinal deviatoric stress with a yield strength estimate from Vaughan (1993). The longitudinal deviatoric stress is problematic for two reasons. First, the various components of the deviatoric stress are not coordinate system invariant and have little physical meaning: a different coordinate system would result in different numbers. It is possible that the authors want to look at, say the components of the traction along the bed (which is well defined) or the largest principle deviatoric stress (which is also well defined). But this raises the more fundamental issue: it is the largest principle Cauchy stress and not the deviatoric stress that controls tensile fracture. And it is clear from the manuscript that the authors are fully focused on tensile basal crevasses. If the authors want to argue that the stresses are sufficient to trigger a tensile basal crevasse then they need to examine the largest principle Cauchy stress. Fortunately, this should be straightforward to compute from the full Stokes model. More problematically for the analysis performed here, for a kilometer thick glacier, the hydrostatic pressure is probably of the order of 10 MPa and may result in a negative (compressive) largest principle Cauchy stress. I should point that this is a common problem when dealing with failure of ice and especially basal crevasses. The most common solution to this problem is to (rather arbitrarily) superpose a hydrostatic pressure associated with water to the largest principle Cauchy stress to simulate the effect of water filled crevasses. This is commonly done and I think the authors could get away with it here if they want. Technically, you can't really do this and the right way to do it is to calculate the Cauchy stress after introducing an infinitely narrow test crack. Doing it the right way, usually results in a compressive stress when using the power-law creep rheology of ice. If the authors go the usual route of*

*superposing a hydrostatic stress field, I do suggest showing the stress with and without water pressure to emphasize that the water pressure is (or is not) critical.*

We have recalculated stresses using the recommended metric of the largest principle stress plus the water pressure, referred to as Effective Principle Stress (EPS) in Benn et al. (2017):

$$\text{EPS} = \sigma_1 + p_w = \frac{\sigma_{xx}+\sigma_{yy}}{2} + \sqrt{\left(\frac{\sigma_{xx}-\sigma_{yy}}{2}\right)^2 + \sigma_{xy}^2} + p_w$$

Unsurprisingly, the resultant stress profiles are similar to those reported in that paper. There is a large concentration of EPS in the location where the longitudinal deviatoric stress ($\tau_{xx}$) peaks. The downstream compressive stress peak and the tensile peak located directly at the bottom corner of the calving front are comparatively greatly diminished in EPS as compared with the same features in $\tau_{xx}$. Figures AC1 and AC2 below compare $\tau_{xx}$ and EPS calculated for the example reported in figure 4 of the original submission.

[Figure]

**Fig AC1.** Comparison of $\tau_{xx}$ (left) and EPS calculated for the geometry shown in Figure 4. of the original submission. Units of stress in MPa. The spatial scale is the same as Figure 4. of the original submission.

[Figure]

**Fig AC2.** Comparison of $\tau_{xx}$ and EPS along the basal boundary, for the same geometry as in Figure AC1.

As the reviewer points out, the ice hydrostatic pressure is of order ~10MPa at the bed and therefore without the addition of the water pressure, the largest principle stress $\sigma_1$ is negative everywhere. It was suggested that we add a plot to show that the water pressure is critical to produce a positive stress. We feel that this would be unnecessary and not particularly instructive, since $p_w$ has a constant value of 9.1MPa along the bed. However we will add a sentence to make it clear that the addition of $p_w$ is critical to our analysis.

The reviewer suggests that the more correct way to carry out the analysis would be to introduce test cracks into the geometry and look at the crack tip stress. We have looked at this (Fig. AC3), but this problem has the stress tend to infinity approaching the crack tip, so that although our simulations do indicate a substantial stress around the tip, the numerical solutions do no converge with mesh resolution.

[Figure]

**Fig AC3.** Demonstration of the dependence of the crack tip stress upon the mesh spacing, with increasing resolution from left to right. The crack is 1m wide at the base, 5m high and located at the position of maximum basal EPS. In each figure, the maximum EPS on the colour bar corresponds to the value of EPS at the crack tip.

One approach to avoid this issue and test the likelihood of the crack to grow or stagnate would be to use the methods of linear elastic fracture mechanics (LEFM) to calculate a stress intensity factor for the crack tip (e.g. van der Veen, 1998; Krug et al., 2014).

We feel that following this method would entail a significant extra addition to the paper at this stage. Therefore our preferred choice is to use the EPS metric. Using this metric, the substantial growth in concentrated EPS following the geometric perturbation would lead to the formation of a crevasse at the same location as the original deviatoric stress metric. Other similar modelling studies (e.g. Nick et al., 2010; Todd et al., 2014) apply the Nye zero stress criterion (Nye, 1957) to calculate the depth of crevasses. Although we do not calculate crevasse depths, we will refer to this criterion so relate the high stresses to locations where crevasses will form.

The change of reported stress metric will require numerous minor changes to figures and text. **Figures 3, 4, 5, 7, 8 and 9 will be revised with the EPS metric replacing the longitudinal deviatoric stress. An additional description of the EPS metric will be added to section 2.1. Minor changes to the text corresponding to the updated figures will be made in sections 3, 3.1 and 4 where appropriate. References to Vaughan (1993) will be removed. A reference to Nye (1957) will be added in section 3.**

2. *Another aspect of the analysis that is somewhat problematic is that the authors are comparing their stress metric to the yield strength estimated by Vaughan (1993). My understanding, however, is that Vaughan (1993) examined various yield strength envelopes,*

*finding that the Von Mises stress envelope provided the best fit to the observations. The Von Mises yield criterion, however, is only equivalent to tensile failure in uniaxial loading, which is not the case for the model considered here. Recalling that the second deviatoric stress invariant invariant is proportional to the Von Mises stress, what I suggest is that in addition to the largest principle Cauchy stress, the authors also consider showing the second deviatoric stress invariant as an additional stress metric. This stress metric can be more directly compared with Vaughan's estimated yield strength. Note that in two dimensions, the second effective deviatoric stress invariant is equal to the maximum shear stress and thus the failure mechanism predicted by this envelope would be shear, rather than purely tensile failure and, if the authors go this route, the authors will need to be careful to point this out. Although we speculated that shear failure is important for tall calving cliff in Bassis and Walker (2012), I'm not aware of any strong observational evidence supporting shear failure in calving so the authors may want to take this suggestion under advisement as the broader community has doubts about the viability of shear failure.*

As discussed in our response to point 1 above, this issue is avoided as a result of switching to use of the EPS stress metric.

3. *Another minor point is that the authors convincingly argue that time scales they are interested are short compared to the time scales of flow and thus they can ignore the effect of ice flow on their experiments. However, if the time scale of interest is short compared to the time scale of flow, then this would seem to imply that an elastic rheology would be appropriate. This is surprising to most, but the elastic stress can be quite different from the viscous stress and this is primarily a consequence of the non-linearity in the creep flow law used.*

We are pleased that the reviewer finds our arguments regarding timescales convincing. We agree with this minor point that the stress should include an elastic component, though we suspect that the boundary conditions are the key factor, rather than nonlinearity. Here we have neglected elastic forces along the lines of (e.g. Benn et al., 2017; O'Leary and Christofferson, 2013) but accept that inclusion would change our results quantitatively. We do still expect a substantial increase in stress around the grounding line, since the force formerly acting on the bed downstream must be transferred upstream.

4. *The authors should be a little bit careful when discussing sliding laws, water pressure and the stress regime because glaciers are actually three-dimensional with bumps in the bed. In three-dimensions, these bumps play a pretty big role in controlling the stress transmission upstream because portions of the calving front maybe well grounded whilst other portions are close to flotation.*

**We will add a paragraph to the discussion along the lines of the following:**

"The reader should note that our model geometry is highly idealised. In reality, glacier beds are highly non-uniform, with variations in geometry, water and overburden pressure. Bedrock bumps therefore play an important role in controlling the stress transmission upstream. It is plausible that these variations could result in basal stress concentrations of a similar magnitude to the mechanism discussed here."

**References**

Bassis, J. N. and Walker, C. C.: Upper and lower limits on the stability of calving glaciers from the yield strength en- velope of ice, P. Roy. Soc. Lond. A Mat., 468, 913–931, https://doi.org/10.1098/rspa.2011.0422, 2012.

Benn, D. I., Aström, J., Todd, J., Nick, F. M., Hulton, N. R., and Luckman, A.: Melt-undercutting and buoyancy-driven calving from tidewater glaciers: new insights from discrete element and continuum model simulations, J. Glaciol., 63, 691–702, 2017.

Krug, J., Weiss, J., Gagliardini, O., and Durand, G.: Combining damage and fracture mechanics to model calving, The Cryosphere, 8, 2101–2117, https://doi.org/10.5194/tc-8-2101- 2014, 2014.

Nick, F. M., van der Veen, C. J., Vieli, A. and Benn, D. I.: A physically based calving model applied to marine outlet glaciers and implications for the glacier dynamics, Journal of Glaciology, 56, 781–794, doi:10.3189/002214310794457344, 2010.

Nye, J. F.: The distribution of stress and velocity in glaciers and ice-sheets, P. Roy. Soc. Lond. A Mat., 239, 113–133, 1957.

O'Leary, M. and Christoffersen, P.: Calving on tidewater glaciers amplified by submarine frontal melting, The Cryosphere, 7, 119–128, doi:10.5194/tc-7-119-2013, 2013.

Todd, J. and Christoffersen, P.: Are seasonal calving dynamics forced by buttressing from ice mélange or undercutting by melting? Outcomes from full-Stokes simulations of Store Gletscher, West Greenland. The Cryosphere, 8, 2353–2365, doi:10.5194/tc-8-2353-2014, 2014

van der Veen, C. J.: Fracture mechanics approach to penetration of bottom crevasses on glaciers, Cold Regions Science and Technology, 27, 213-223, doi:10.1016/S0165-232X(98)00006-8, 1998.

Vaughan, D. G.: Relating the occurrence of crevasses to surface strain rates, Journal of Glaciology, 39, 255-266, doi:10.1016/0148-9062(94)90888-5, 1993.

---

## Author Response (AR1)

**Author's response to referee comments on "Buoyant forces promote tidewater glacier iceberg calving through large basal stress concentrations"**

*Authors*: Trevers, M., Payne, A. J., Cornford, S. L., Moon, T.

The Cryosphere Discuss., https://doi.org/10.5194/tc-2018-212

We would like to thank both reviewers for their constructive and insightful criticisms and comments on our manuscript. These are taken account of in our revised manuscript. Here we reply to the more important criticisms.

Below, we respond to all comments from both reviews. Reviewer comments, numbered chronologically, are in italics, and the author response in plain text. Details of all changes to the manuscript to address comments are highlighted in bold text. References to the location in the manuscript of changes refer to the revised manuscript *without* markup displayed

**Reviewer #1**

1. *I'm sorry, but so much of the discussion in the manuscript depends on un-referenced and un-explained "references to previous work" that I think that the manuscript needs to be revised significantly. The specific places where the discussion and explanations are inadequate are listed below.*

We agree that the exposition of the proposed mechanism inducing the stress concentrations is not entirely clear, and that we've also used some jargon in some places that hasn't been properly defined for the reader. Rewording and some additional exposition will be added to the manuscript, with detail of the changes given in response to specific comments below.

**No changes made in response to this specific comment. Changes that account for this general comment are detailed in response to comment #11**

2. *Other than the expository problem with the manuscript. I find the science compelling and well done. The work is creative and important in the study of iceberg calving mechanisms. Here's the stuff that needs attention (in my estimation):*

Thank you very much for the generous remark, we're glad that you found the work compelling!

**No changes made in response to this comment.**

3. *I wonder if the title really does justice. . . the paper is about bending moments (viscous and plastic bodies have bending moments too!) generated by geometry changes at the ice front due to ice/ocean and ice/atmosphere and ice/wave interactions. . . the present title could be misunderstood to represent "same old basal shear stress" stuff. . ..*

The paper is not specifically about bending moments but rather about forces generated at the bed instantaneously at the onset of bending in response to the perturbation. Some further explanation is required on this for clarity (see the response to the next comment), but as such, we do not intend to change the title.

**No changes made in response to this specific comment.**

4. *line 13 - would it be more accurate to say "viscous bending moment" (remember you can bend a beam viscously and elastically and viscoelastically) leading to high tensile stress concentration at the bed. . . instead of stresses at the ice-bed interface? Who cares what the stresses are at the interface if the ice is actually in a state of bending induced fracture?*

As above, we don't attribute these stresses to a bending moment. We attribute these longitudinal stresses as those required to balance the abrupt decrease in basal shear across the grounding zone.

**We have added a new paragraph starting at page 8, line 4 to clarify and justify our interpretation of these stresses.**

> 5. *line 29 - Would this be a place to add a reference to Weertman?*

**A reference to Weertman (1957) for the Weertman sliding law has been added to page 4, line 7. Lines 7-9 of page 4 have been slightly reworded to account for this addition.**

> 6. *line 18 page 2 - The rotation should be indicated as "bottom out".*

**The description has been changed to refer to "bottom-out" calving events in page 2 line 10, page 2 line 19 and the caption for figure 1.**

> 7. *line 8 page 3 - If I were to be pedantic, I would say that a reference should be given for "Stokes equations" (in actuality, Stokes was prolific and probably has many equations associated with his name). Ditto for "Glen's flow law". . . a reference should be given.*

**A reference to Gagliardini et al. (2013) which solves to same Stokes equations has been provided on page 3 line 9. A reference to Glen (1958) has been provided on page 3 line 13.**

> 8. *line 1 page 4 - Is Cauchy stress the same as deviatoric stress?*

The deviatoric stress indicates the deviation from the Cauchy (or full) stress, i.e. it negates the average pressure term. For our ~1km thick glacier, the pressure will be ~10MPa everywhere along the bed and therefore the Cauchy stress, ignoring additional water pressure, will be compressive everywhere.

**This detail has been clarified in the paper with an adjustment to equation 2, and changes to the text in page 3 lines 10-12. Also, removal of text in page 4, line 1 which was rendered redundant. The identity matrix was added to table 1.**

> 9. *line 9 page 5 - Just out of curiosity why are 191 and 644 meters so precisely known as to be significant to the single meter? Can the authors tell us what would happen if the numbers were 192 and 643?*

These numbers are part of the model output and the precision is fairly arbitrary. The precision of these numbers is not important to our argument, and is in a small part determined by the mesh resolution (i.e. the peak occurs at a mesh node). Since it is not possible to identify locations to the nearest meter in the figures provided, we will change to reporting them as "approximately 190 m" and similarly for other such numbers.

**Changes made to: page 5, line 23; page 5, line 25; page 6, line 2; figure captions for figures 4, 5, 8, 9, 10.**

> 10. *section 3.1 - What is the a priori reason to expect water pressure to be significantly important in the problem? is it for promoting fracture propagation or is it for lubricating the base?*

The significance is that the mechanism we suggest relies on the abrupt reduction in basal shear stress where the ice ungrounds to produce the longitudinal stresses to balance this. If we apply a sliding law where the basal shear stress reduces gradually as a function of effective pressure, we wouldn't expect the same result.

**A sentence has been added to page 7, line 4 to clarify the expectation of different results with a different sliding law.**

11. *line 4 page 7 - What is "Weertman-like"???? This seems to come in out of the blue. . . Weertman published hundreds of papers in his life, what is referred to here?*

   *line 1 page 9 - Notch-triggered rotation mechanism was shown to be irrelevant under the full School regime". . . Readers will get confused here, because "full School regime" is a weak form of jargon that doesn't really convey the precise ideas (regardless of whether they are published in Schoof. . . my hunch is that the authors have a different meaning, i.e., an interpretation that they ascribe to the term "full School regime")*

   *line 12 and 13 page 9 - "Weertman . . . regime"???? Weertman-like stress peak???? What is this, and why the name Weertman???? Seems like citations and explanations are required. The discussion is flawed because it relies on readers having prior knowledge of what a "Weertman regime" is...*

I've grouped these comments together because they all refer to the same thing. We agree with the reviewer that we've used jargon, the meaning of which has been poorly conveyed to the reader. When referring to "regimes", we meant the use of the Weertman or Schoof sliding law. The "full" Schoof regime referred to the use of the Schoof sliding law with full hydrostatic water pressure applied along the ice-bed interface. "Weertman-like stress peak" referred to stress concentrations that have a similar profile to those produced using the Weertman sliding law.

**Numerous minor changes to the text have been made to clarify this: Page 7, line 27; page 10, line 7; page 10, lines 18-20.**

**Reviewer #2**

1. *There is one major issue with the analysis, which is that the authors compare longitudinal deviatoric stress with a yield strength estimate from Vaughan (1993). The longitudinal deviatoric stress is problematic for two reasons. First, the various components of the deviatoric stress are not coordinate system invariant and have little physical meaning: a different coordinate system would result in different numbers. It is possible that the authors want to look at, say the components of the traction along the bed (which is well defined) or the largest principle deviatoric stress (which is also well defined). But this raises the more fundamental issue: it is the largest principle Cauchy stress and not the deviatoric stress that controls tensile fracture. And it is clear from the manuscript that the authors are fully focused on tensile basal crevasses. If the authors want to argue that the stresses are sufficient to trigger a tensile basal crevasse then they need to examine the largest principle Cauchy stress. Fortunately, this should be straightforward to compute from the full Stokes model. More problematically for the analysis performed here, for a kilometer thick glacier, the hydrostatic pressure is probably of the order of 10 MPa and may result in a negative (compressive) largest principle Cauchy stress. I should point that this is a common problem when dealing with failure of ice and especially basal crevasses. The most common solution to this problem is to (rather arbitrarily) superpose a hydrostatic pressure associated with water to the largest principle Cauchy stress to simulate the effect of water filled crevasses. This is commonly done and I think the authors could get away with it here if they want. Technically, you can't really do this and the right way to do it is to calculate the Cauchy stress after introducing an infinitely narrow test crack. Doing it the right way, usually results in a compressive stress when using the power-law creep rheology of ice. If the authors go the usual route of superposing a hydrostatic stress field, I do suggest showing the stress with and without water pressure to emphasize that the water pressure is (or is not) critical.*

We have recalculated stresses using the recommended metric of the largest principle stress plus the water pressure, referred to as Effective Principle Stress (EPS) in Benn et al. (2017):

$$\text{EPS} = \sigma_1 + p_w = \frac{\sigma_{xx} + \sigma_{yy}}{2} + \sqrt{\left(\frac{\sigma_{xx} - \sigma_{yy}}{2}\right)^2 + {\sigma_{xy}}^2} + p_w$$

Unsurprisingly, the resultant stress profiles are similar to those reported in that paper. There is a large concentration of EPS in the location where the longitudinal deviatoric stress ($\tau_{xx}$) peaks. The downstream compressive stress peak and the tensile peak located directly at the bottom corner of the calving front are comparatively greatly diminished in EPS as compared with the same features in $\tau_{xx}$. Figures AC1 and AC2 below compare $\tau_{xx}$ and EPS calculated for the example reported in figure 4 of the original submission.

[Figure]

**Fig AC1.** Comparison of $\tau_{xx}$ (left) and EPS calculated for the geometry shown in Figure 4. of the original submission. Units of stress in MPa. The spatial scale is the same as Figure 4. of the original submission.

[Figure]

**Fig AC2.** Comparison of $\tau_{xx}$ and EPS along the basal boundary, for the same geometry as in Figure AC1.

As the reviewer points out, the ice hydrostatic pressure is of order ~10MPa at the bed and therefore without the addition of the water pressure, the largest principle stress $\sigma_1$ is negative everywhere. It was suggested that we add a plot to show that the water pressure is critical to produce a positive stress. We feel that this would be unnecessary and not particularly instructive, since $p_w$ has a

constant value of 9.1MPa along the bed. However we will add a sentence to make it clear that the addition of $p_w$ is critical to our analysis.

The reviewer suggests that the more correct way to carry out the analysis would be to introduce test cracks into the geometry and look at the crack tip stress. We have looked at this (Fig. AC3), but this problem has the stress tend to infinity approaching the crack tip, so that although our simulations do indicate a substantial stress around the tip, the numerical solutions do no converge with mesh resolution.

[Figure]

**Fig AC3.** Demonstration of the dependence of the crack tip stress upon the mesh spacing, with increasing resolution from left to right. The crack is 1m wide at the base, 5m high and located at the position of maximum basal EPS. In each figure, the maximum EPS on the colour bar corresponds to the value of EPS at the crack tip.

One approach to avoid this issue and test the likelihood of the crack to grow or stagnate would be to use the methods of linear elastic fracture mechanics (LEFM) to calculate a stress intensity factor for the crack tip (e.g. van der Veen, 1998; Krug et al., 2014).

We feel that following this method would entail a significant extra addition to the paper at this stage. Therefore our preferred choice is to use the EPS metric. Using this metric, the substantial growth in concentrated EPS following the geometric perturbation would lead to the formation of a crevasse at the same location as the original deviatoric stress metric. Other similar modelling studies (e.g. Nick et al., 2010; Todd et al., 2014) apply the Nye zero stress criterion (Nye, 1957) to calculate the depth of crevasses. Although we do not calculate crevasse depths, we will refer to this criterion so relate the high stresses to locations where crevasses will form.

**The following changes were made to account for this:**

- **Addition of section 2.3**
- **Figure 3 and caption**
- **Figure 4 and caption**
- **Figure 5 and caption**
- **Figure 7 and caption**
- **Figure 8 replotted as two figures, figure 8 and figure 9. This is to account for the large shift in the vertical scale when switching to the EPS metric.**
- **Figure 9 replotted and caption updated (and changed to figure 10)**
- **Changed "longitudinal deviatoric stress" to "EPS" at page 5, line 20.**
- **Removal of the reference to a tensile strength from Vaughan (1993) at page 5, line 21.**
- **Removal of the reference to tensile strength at page 5, line 25.**
- **Added sentence at page 5, line 28.**

- **Updated text at page 7, lines 12 – 15 to account for changed figures and the updated results.**
- **Changed text at page 10, line 12 to account for the updated results using EPS.**
- **EPS and $\sigma_1$ added to table 1.**

2. *Another aspect of the analysis that is somewhat problematic is that the authors are comparing their stress metric to the yield strength estimated by Vaughan (1993). My understanding, however, is that Vaughan (1993) examined various yield strength envelopes, finding that the Von Mises stress envelope provided the best fit to the observations. The Von Mises yield criterion, however, is only equivalent to tensile failure in uniaxial loading, which is not the case for the model considered here. Recalling that the second deviatoric stress invariant invariant is proportional to the Von Mises stress, what I suggest is that in addition to the largest principle Cauchy stress, the authors also consider showing the second deviatoric stress invariant as an additional stress metric. This stress metric can be more directly compared with Vaughan's estimated yield strength. Note that in two dimensions, the second effective deviatoric stress invariant is equal to the maximum shear stress and thus the failure mechanism predicted by this envelope would be shear, rather than purely tensile failure and, if the authors go this route, the authors will need to be careful to point this out. Although we speculated that shear failure is important for tall calving cliff in Bassis and Walker (2012), I'm not aware of any strong observational evidence supporting shear failure in calving so the authors may want to take this suggestion under advisement as the broader community has doubts about the viability of shear failure.*

As discussed in our response to point 1 above, this issue is avoided as a result of switching to use of the EPS stress metric.

**No changes were made in response to this specific comment. Changes already detailed in response to comment #1 remove the Vaughan (1993) tensile strength as a crevassing criterion, but it is still referenced in section 2.3 to justify our interpretation of the results.**

3. *Another minor point is that the authors convincingly argue that time scales they are interested are short compared to the time scales of flow and thus they can ignore the effect of ice flow on their experiments. However, if the time scale of interest is short compared to the time scale of flow, then this would seem to imply that an elastic rheology would be appropriate. This is surprising to most, but the elastic stress can be quite different from the viscous stress and this is primarily a consequence of the non-linearity in the creep flow law used.*

We are pleased that the reviewer finds our arguments regarding timescales convincing. We agree with this minor point that the stress should include an elastic component, though we suspect that the boundary conditions are the key factor, rather than nonlinearity. Here we have neglected elastic forces along the lines of (e.g. Benn et al., 2017; O'Leary and Christofferson, 2013) but accept that inclusion would change our results quantitatively. We do still expect a substantial increase in stress around the grounding line, since the force formerly acting on the bed downstream must be transferred upstream.

**No changes were made in response to this comment.**

4. *The authors should be a little bit careful when discussing sliding laws, water pressure and the stress regime because glaciers are actually three-dimensional with bumps in the bed. In three-dimensions, these bumps play a pretty big role in controlling the stress transmission*

*upstream because portions of the calving front maybe well grounded whilst other portions are close to flotation.*

**A new paragraph has been added at page 10, line 2 to communicate this point to the reader.**

**Additional changes**

The authors made some minor further changes in order to correct minor mistakes, clarify meaning or improve the flow of the text, which were not based on recommendations by the reviewers.

- **Changed "mass balance of the Greenland Ice Sheet" to "Greenland Ice Sheet mass balance", page 1 line 21.**
- **Added Greenlandic name of JI, page 1 line 24.**
- **Changed "superbuoyancy" to "super-buoyancy", page 2 line 7.**
- **Added an extra final line to paragraph, page 2 line 13.**
- **Removed "further we", page 2, line 25.**
- **Removed "however", page 2, line 31.**
- **Added "when not present", page 2 line 22.**
- **Added "applied here", page 9 line 5.**
- **Changed "suggests" to "suggested", page 9 line 9.**
- **Changed "trigger" to "induce", page 9 line 16.**
- For the sake of consistency, "basal drag" has been replaced with "basal shear stress" since these two terms were being used interchangeably. **Changes made at page 4, line 9; page 4, line 15; page 5, line 27; page 7, line 31; page 9, line 4; page 9, line 28, table 1.**
- **Updated references to Cuffey and Patterson (2010) to account for differing page numbers.**
- **Updated figure 6 to match format of other figures. Figure content and caption unchanged.**
- **Added "The", to the start of page 10, line 7.**
- **Changed "significantly" to "greatly" to enhance readability, page 10, line 26.**
- **Added "Code availability" section.**
- **Updated "Acknowledgements" section to clarify MT's grant reference number.**
- **Corrected numerous typos in the references.**

[revised manuscript text omitted]

---

## Author Response (AR2)

**Author's response to referee comments on the revised version of "Buoyant forces promote tidewater glacier iceberg calving through large basal stress concentrations"**

Authors: Trevers, M., Payne, A. J., Cornford, S. L., Moon, T.

The Cryosphere Discuss., https://doi.org/10.5194/tc-2018-212

5 We would like to thank Referee #2 for their comments on our revised manuscript. Below we respond to all comments and indicate where corrections and additions have been made to the manuscript in response to these. Referee comments are numbered and in italics, changes to the manuscript are indicated in bold. Location of changes to the manuscript assume no markup is shown.

**Referee #2**

- The Durand et al., (2009) method to solve the contact problem involves introducing a vertical damping force to balance buoyant uplift in portions of the glacier. The reason why a buoyant damping force is needed is to keep the problem well posed and allow closure in the vertical force balance without needing to resolve the inertial terms. However, the method proposed by Durand et al., (2009) usually results in a time step dependent velocity field. This suggests two concerns:
- The authors should state what time step was used in solving the contact problem. I apologize if this was included and I missed it or this is not actually what the authors are going;
   The authors should characterize the time-step sensitivity of the strain rate and stress field. In my experience, for relatively small changes between time steps, the strain rate field, which is related to the symmetric gradient of the velocity field is only weakly time step independent. However, the strong
   change in geometry associated with the block removal is larger than anything that I have examined. If there is no time step sensitivity, then the authors can merely state this quickly to avoid any future confusion. I don't anticipate that this will be a problem, but even if there is some time step sensitivity, then simply documenting it would be sufficient from my perspective. It isn't my goal to act as some kind of gatekeeper and the idea that the decrease in longitudinal stress could promote calving still merits
- 25 consideration in the literature.

30

In the prognostic case, the solution of the non-linear viscous flow relation, nonlinear friction law and the solution-dependent position of the grounding line are treated together iteratively during the nonlinear iterations of the Stokes equations, assuming a fixed grounding line calculated from the previous timestep. Once the solution has converged slightly then the new grounding line is applied and the solution calculated using this. This new grounding line is then applied at the start of the nonlinear iterations at the next timestep.

In our diagnostic case there is no timestepping and the geometry does not evolve, but the location of the grounding is determined by the water pressure at the ice base. Instead of evolving the solution iteratively with each timestep, we use 5 steady-state iterations of the non-linear loop with the geometry kept fixed (although the calculated grounding line position evolves with the water pressure).

Beyond 5 steady-state iterations, further changes to the grounding line position and the stress and velocity fields are insignificant.

**Additional text has been added to the manuscript at page 5, line 2 to detail this.**

- 2) I also have a question about the mechanism resulting in large basal stress concentrations in the Weertman sliding law. The transition from no-slip to slip is known to result in a singularity in the stress field at the slip/no-slip transition. Including a sliding law, like the Weertman sliding law, allows a smooth transition from slowly slipping to freely slipping and this removes the singularity in the stress field. The introduction of the Weertman sliding law essentially regularizes the singularity in the crack problem, but large values of stress are still expected near the transition to free-slip and these values of stress should
- 10 increase with the sliding law coefficient. This looks like what is shown in Figure 5. However, it is often necessary to increase the resolution of a model near the slip transition to fully capture the peak in stress. Is it the slow slip/free slip transition that is generating the large stress concentrations> Are these numerically resolved? Does the magnitude of the stress concentration increase with increasing friction coefficient?
- 15 This also seems inconsistent with Figure 6, which I really don't understand. For small friction coefficients, the ice is essentially freely-slipping so I don't quite understand why the transition to buoyancy decreasing the shear stresses matters in this regime. Shear stress are already negligible and decreasing them further seems like it shouldn't make a difference after a point. Clearly, large basal friction coefficients are more stable and this makes me wonder how much the sensitivity depends on the baseline longitudinal stresses (smaller for high friction and larger for low friction). 20

The basal shear stress is discontinuous across the slow-slip/free-slip transition. It is this discontinuity that generates the large stress concentration. There is only weak sensitivity of the location and magnitude of the stress peak to the mesh resolution. Convergence of the solution with mesh resolution was confirmed by performing a Richardson extrapolation. The magnitude of the stress concentration increases with increasing

friction coefficient (see figure 7). 25

5

30

We used a mesh resolution along the basal boundary of 4m at the calving front increasing linearly to 8m at the rear of the domain. An additional sentence at page 3, line 5 details this.

We are unclear what is being asked in the comment referring to Figure 6.

- 3) Page 4, near line 15. The discussion of observations of fracture strength mixes quite different loading
- regimes. The yield criterion that Vaughan (1993) deduced was based on the Von Mises stress criterion. This is only equal to the tensile stress used by the authors in purely tensile loading. An appropriate comparison would plot the Von Mises stress. Or stick to the Schulson experiments. The experiments in Schulson (2001), specifically include tensile failure experiments, which seem more in line with largest principle stress criterion that the authors prefer for failure.
- 35 The sentence relating to Vaughan (1993) has been removed from page 2, line 20, along with the corresponding reference.

4) Equation (9) doesn't provide a sign. I assume that the actual implementation includes multiplying by the sign of the basal velocity or some such additional implementation decision to ensure that friction opposes flow. Mentioning how this is implemented in Elmer/Ice would be nice.

The basal sliding velocity magnitude  $|u_b|$  is included inside the brackets in eqn. 9 to produce the slip coefficient.

- 5 This is then multiplied by the velocity to produce the basal friction. We've included the minus sign because it acts in the opposite direction to the velocity. Eqn. 10 should also include the magnitude of sliding. Eqns 9 and 10 have been updated in the manuscript to correct this.
  - 5) How were the constants in the Coulomb-limited sliding law chosen? Are these determined so that they roughly correspond to the same basal shear stress as the Weertman sliding law?
- 10 The values of  $C_c$  and q are commonly used values for these parameters. The value of  $A_s$  was chosen such that the sliding velocity and basal shear stress roughly matched the Weertman case. This justification is included at page 7, line 14.
  - 6) Page 4 near Line 25, It says the vertical velocity is set to zero when the ice is grounded, but for the contact problem, isn't it the velocity normal to the bed zero?
- 15 Correct. In this instance there is no difference, since the bed is horizontal. However, I've updated the manuscript to reflect this at page 5, lines 1 and 2 to reflect this more correct description.

**Additional changes**

- Eqn 8 changed for consistency with the form of eqn 9.
- Eqn 14 changed for consistency with the form of eqn 9.
- Noted the basal sliding velocity notation at page 4, line 9.
  - Minor changes to make clear that ub and τb are vectors by changing them to bold at page 4, lines 9 and 18, and in table 1.

3

- For consistency, all vector are now shown in italics. Changes made in table 1.
- Corrected a typo in the units of As in table 1.
- Included the missing reference to Gagliardini et al. (2007).
  - Corrected the spelling of "Råback" in the reference to Gagliardini et al. (2013).

30

[revised manuscript text omitted]
 = -\mathcal{C}  \boldsymbol{u}_b  + \boldsymbol{u}_b + \boldsymbol{u}_b + \boldsymbol{u}_b \boldsymbol{u} $                                     |
|----------------------------------------------------------------------------------------------------------------------------------------------------------------------|
| with $\boldsymbol{z}_b$ the basal shear stress, C the Weertman friction coefficient, $\underline{u}_b$ the basal sliding velocity and sliding exponent $m = 1/2$     |
| Values of C range from $10^5$ to $10^8$ Pa m -1/3 s 1/3 , which includes the more realistic range of modelled values of ~ $10^6$ to ~ $10^7$ H |
| m -1/3 s 1/3 determined from surface velocity observations around Greenland outlet glaciers (Lee et al., 2015).                                |

Alternatively, a Coulomb-limited sliding law (Schoof, 2005; Gagliardini et al., 2007) can be applied (referred to as the "Schoof law" from here on in). This law accounts for the effect of water pressure through an effective pressure term  $N = -\sigma_{nn} - P_w$ . Basal shear stress is expressed as

$$\boldsymbol{\tau}_{b} = -C_{c} \cdot N \left( \frac{\chi |\boldsymbol{u}_{b}|^{-n}}{1 + \alpha \chi^{q}} \right)^{\frac{1}{n}} \cdot \boldsymbol{u}_{b}$$
(9)

where

$$\chi = \frac{|\boldsymbol{u}_b|\boldsymbol{u}}{C_c{}^n N^n A}$$

and

 $\alpha = \frac{(q-1)^{q-1}}{a^q}.$

[revised manuscript text omitted]
'                |                           | К                                      |
| Velocity tensor               | μ                 |                           | m s -1                      |

| Basal sliding velocity                                        | u b |       | m s-1 |                         |
|---------------------------------------------------------------|------------------------------|-------|-------------------------|-------------------------|
| Glacier surface gradient                                      | α                            | 2 - 5 | 0                       |                         |
| Strain rate tensor                                            | Ė                            |       | $s^{-1}$                |                         |
| Square of $2^{nd}$ invariant of $\dot{\boldsymbol{\epsilon}}$ | $\dot{\epsilon}_e^2$         |       | s -2         |                         |
| Effective viscosity                                           | μ                            |       | Pa s                    |                         |
| Ice density                                                   | $ ho_{ m i}$                 | 918   | kg m -3      |                         |
| Water density                                                 | $ ho_{ m w}$                 | 1028  | kg m -3      |                         |
| Cauchy stress tensor                                          | ø                            |       | Ра                      | Formatted: Font: Italic |
| Largest principal Cauchy stress                               | $\sigma_1$                   |       | Pa                      |                         |
| Deviatoric stress tensor                                      | Ţ                            |       | Ра                      | Formatted: Font: Italic |
| Basal shear stress                                            | T b                   |       | Pa                      | Formatted: Font: Bold   |
|                                                               |                              |       |                         |                         |

Table 1. Symbols and values of physical and numerical constants and parameters used in this study.

| Experiment | Hydraulic connectivity | $l_{n}\left(\mathrm{m}\right)$ | x 1 (m) | x 0 (m) |
|------------|------------------------|--------------------------------|---------------------------|---------------------------|
| F0         | Full                   | 0                              | 10000                     | 10000                     |
| F100       | Full                   | 100                            | 10000                     | 10000                     |
| Z0         | Zero                   | 0                              | 0                         | 0                         |
| Z100       | Zero                   | 100                            | 0                         | 0                         |
| P0         | Partial                | 0                              | 0                         | 10000                     |
| P100       | Partial                | 100                            | 0                         | 10000                     |

Table 2. Hydraulic connectivity along the ice-bed interface for experiments using the Schoof law. Water pressure is 100% of the full hydrostatic pressure (Eq. 7) downstream of position  $x_1$ . Between  $x_1$  and  $x_0$  water pressure reduces linearly to 0%.

Figure 1. Proposed calving mechanism. (a) Lightly grounded terminus of a tidewater glacier with approximate dimensions of e.g. Jakobshavn Isbræ. (b) A weakness develops in the subaerial section of the front due to (e.g.) undercutting by a wave-cut notch at the waterline. (c) A small subaerial calving event rapidly increases the buoyant load, causing the terminus to tend to lift and rotate. 5 Basal crevases open and propagate rapidly upwards. (d) Full-depth crevassing results in a large, bottom-out calving event. The long-term calving rate is driven by the notch melt rate but is amplified by an unconstrained factor.

Figure 2. Example mesh and boundary conditions (not to scale). Mesh resolution increases close to the calving front and basal boundaries. Symbols: normal stress  $\sigma_{nn}$ , shear stress  $\sigma_{nt}$ .